# Inactivation of SARS-CoV-2 virus in saliva using a guanidium based transport medium suitable for RT-PCR diagnostic assays

Sukalyani Banik[1], Kaheerman Saibire[1], Shraddha Suryavanshi[1], Glenn Johns[2], Soumitesh Chakravorty[2], Robert Kwiatkowski[2], David Alland[1]☉*, Padmapriya P. Banada[1]☉*

1 Department of Medicine, Center for Emerging Pathogens, Rutgers-New Jersey Medical School, Newark, New Jersey, United States of America, 2 Cepheid, Sunnyvale, California, United States of America

☉ These authors contributed equally to this work.
* allandda@njms.rutgers.edu (DA); priya.banada@rutgers.edu (PPB)

**Data Availability Statement:** All relevant data are within the paper.

## Abstract

### Background

Upper respiratory samples used to test for SARS-CoV-2 virus may be infectious and present a hazard during transport and testing. A buffer with the ability to inactivate SARS-CoV-2 at the time of sample collection could simplify and expand testing for COVID-19 to non-conventional settings.

### Methods

We evaluated a guanidium thiocyanate-based buffer, eNAT™ (Copan) as a possible transport and inactivation medium for downstream Reverse Transcriptase-Polymerase Chain Reaction (RT-PCR) testing to detect SARS-CoV-2. Inactivation of SARS-CoV-2 USA-WA1/2020 in eNAT and in diluted saliva was studied at different incubation times. The stability of viral RNA in eNAT was also evaluated for up to 7 days at room temperature (28˚C), refrigerated conditions (4˚C) and at 35˚C.

### Results

SARS-COV-2 virus spiked directly in eNAT could be inactivated at >5.6 $\log_{10}$ PFU/ml within a minute of incubation. When saliva was diluted 1:1 in eNAT, no cytopathic effect (CPE) on VeroE6 cells was observed, although SARS-CoV-2 RNA could be detected even after 30 min incubation and after two cell culture passages. A 1:2 (saliva:eNAT) dilution abrogated both CPE and detectable viral RNA after as little as 5 min incubation in eNAT. SARS-CoV-2 RNA from virus spiked at 5X the limit of detection remained positive up to 7 days of incubation in all tested conditions.

### Conclusion

eNAT and similar guanidinium thiocyanate-based media may be of value for transport, stabilization, and processing of clinical samples for RT-PCR based SARS-CoV-2 detection.

**Funding:** This study was funded by the National Institute of Allergy and Infectious Diseases of the National Institutes of Health under award number R01 AI131617 and a research grant provided by Cepheid LLC. -DA received the funding from both NIH/NIAID and Cepheid - NIH R01 AI131617. Cepheid supplied the resources including the GeneXpert instrument and the Xpert Xpress SARS-CoV-2 cartridges - National Institute of Allergy and Infectious Diseases of the National Institutes of Health and Cepheid, LLC. - https://www.niaid.nih. gov/about/visitor-information and https://www. cepheid.com/ Yes. Cepheid employees GJ, SC and RK were involved in the study design, review of the data and the final draft of the manuscript.

**Competing interests:** R.K. G.J, and S.C. are employees of Cepheid Inc., which sells the Xpert Xpress SARS-CoV-2 test. D.A. receives research support and royalty payments from Cepheid. This does not alter our adherence to PLOS ONE policies on sharing data and materials.

## Introduction

The novel coronavirus SARS-CoV-2 (Severe Acute Respiratory Syndrome Coronavirus 2), the causative agent of Coronavirus induced disease 2019 (COVID19), emerged in Wuhan, China at the end of 2019 and has since infected almost one hundred fifty million people worldwide, causing a global pandemic [1–3]. Simple and rapid methods to detect SARS-CoV-2 have the potential to aid in controlling the spread of COVID-19. However, clinical samples obtained for SARS-CoV-2 testing can present a biohazard during transport or processing in a testing laboratory [4]. Clinical samples obtained at home or in remote locations can leak during transport to a testing laboratory, presenting a biohazard anywhere along this transport chain. Furthermore, the CDC recommends that tests for SARS-CoV-2 should be performed inside a biosafety cabinet in a BSL-2 laboratory setting (https://www.cdc.gov/coronavirus/2019-ncov/ lab/lab-biosafety-guidelines.html) to ensure worker safety. These strict testing requirements limit the locations where SARS-CoV-2 assays can be safely performed. A transport buffer that inactivates SARS-CoV-2 while remaining suitable for RT-PCR detection assays would mitigate these safety threats and simplify widespread COVID-19 testing.

Many of the physical (heat, ultraviolet light, gamma irradiation) and chemical (detergents, alcohol) methods used for viral inactivation or RNA extraction are either unsuitable for creating a safe transport media [5–9] or require additional downstream sample processing that would complicate rapid point of care COVID-19 tests [10]. Buffers and reagents containing guanidium hydrochloride and guanidinium thiocyanate have shown to inactivate SARS-CoV-2, and these reagents can be used for RT-PCR applications [5,9,11,12]. The eNAT (Copan Diagnostics, Murrieta, CA) is a guanidium thiocyanate based medium designed to be used for specimen collection and transport as it can stabilize nucleic acids for prolonged periods of time (https://www.copanusa.com/sample-collection-transport-processing/enat/) [13]. In this study, we evaluated the effectiveness of this media for inactivating SARS-CoV-2 in cell culture supernatants and virus-spiked saliva matrix and its ability to be used to detect SARS-CoV-2 in the point-of-care Xpert Xpress SARS-CoV-2 assay. We also studied the stability of viral RNA in presence of a clinical matrix added to this medium. The findings of this study can help simplify sample collection, transport, and COVID-19 diagnostic workflows, potentially increasing access to testing and reducing the time to testing results.

## Materials & methods

### Ethical considerations

Saliva samples were collected from healthy volunteers in a sterile container as approved by Rutgers Institutional Review Board, IRB# 2020001786 and tested for SARS-CoV-2 using Xpert Xpress SARS-CoV-2 test (Cepheid, Sunnyvale, CA). Only confirmed SARS-CoV-2 negative saliva samples were used for this study and a signed consent was obtained from all volunteers.

### Cell and viral culture

Vero E6 cells (ATCC CRL-1586) were maintained in the Dulbecco's Modified Eagle Medium (DMEM, Gibco, Thermo Fisher Scientific, Waltham, MA) supplemented with 10% heat-inactivated fetal bovine serum (FBS, Gibco, Thermo Fisher Scientific, Waltham, MA) and 100 units/ml of penicillin/streptomycin (Gibco, Thermo Fisher Scientific, Waltham, MA) at 37˚C in the presence of 5% $CO_2$. SARS-CoV-2 strain USA-WA1/2020 was obtained from the World Reference Collection for Emerging Viruses and Arboviruses (WRECVA) at the University of Texas Medical Branch (UTMB, Galveston, TX) and further propagated in Biosafety level 3 (BSL3) laboratory at the Regional Biocontainment Laboratory (RBL),

Rutgers New Jersey Medical School, Newark, NJ. All cell lines and virus cultures were maintained at 37°C in the presence of 5% $CO_2$ unless otherwise mentioned. All experiments in this study were performed inside a biosafety cabinet within a BSL3 containment facility. To generate working virus stocks, Vero E6 cells were infected with a multiplicity of infection (MOI) of 0.01 in DMEM supplemented with 2% FBS. Cells were harvested at 72 hours post-infection, supernatants were collected and centrifuged for 10 min at 1,000×g, aliquoted and stored at -80°C.

## Virus quantitation

SARS-CoV-2 virus titers were determined using both plaque assays and the 50% tissue culture infective dose (TCID50) method. Plaque assays were performed following standard procedures [14–16] with some modifications. Briefly, Vero E6 cells were seeded into 6-well plates ($5×10^5$ cells/well) 24 hours before infection. Ten-fold serial dilutions of virus stock were added onto wells (400μl/well) and incubated for 1 hour at 37°C with intermittent shaking every 15 min to prevent the cell monolayers from drying. After 1 hour of virus adsorption, 4ml of 0.8% agarose in DMEM supplemented with 4% FBS was added to each well and incubated for 2–3 days at 37°C/$CO_2$ incubator. The plaques were developed by fixing the cells with 4% formaldehyde in PBS for 1h at room temperature (RT). The agarose plug was removed before staining with 0.2% crystal violet (in 20% ethanol for 15 min at RT). The wells were washed with tap water, dried and the plaques were counted. TCID50 assays were performed by seeding 96-well plates with Vero E6 cells ($2×10^4$ cells/well) the day before the assay. Ten-fold serial dilutions of virus stock were added onto wells (100μl/well) and incubated for 7–10 days at 37°C/$CO_2$ incubator. The plates were observed for presence of cytopathic effect (CPE) every day. TCID50 titers were calculated using the Reed and Muench method [17].

## Removal of cytotoxicity

Four different approaches were explored to remove cytotoxicity induced by the transport media in the absence of the virus. Pierce 4ml Detergent Removal Spin Column (DRSC, Thermo Fisher Scientific, Waltham, MA), Amicon Ultra 4ml 100KDa centrifugal filter units (Millipore Sigma, St. Louis, MO), PD 10 desalting spin columns (Millipore Sigma, St. Louis, MO) and Slide-A-Lyzer G2 dialysis cassette 10K MWCO (Thermo Fisher Scientific, Waltham, MA) were prepared according to each manufacturer's instructions. Two ml of eNAT (Copan Diagnostics, Murrieta, CA) was added to 100 μl of DMEM without the virus, mixed and then processed using each method. Both PD 10 columns and Slide-A-Lyzer G2 dialysis cassettes were equilibrated with PBS before adding samples. For the Slide-A-Lyzer G2 dialysis cassette, the 2.1 ml samples were added, and the cassette was dialyzed overnight at room temperature in PBS with a total of five buffer changes. For Amicon ultra filters, the 2.1 ml samples were added, centrifuged at 4,000 ×g for 10 min and washed three times with 2ml of PBS. For PD 10 columns, the 2.1 ml samples were added, and the eluate was collected by centrifugation at 1000 ×g for 2 min. For the Pierce DRSC columns, the 2.1 ml samples were split into two 1.05 ml samples that were each added to one of two columns, incubated for 2 min at room temperature and then eluted by centrifugation at 1000 ×g for 2 min. Four hundred microliters of the samples processed by each method were then added to Vero E6 cells in 6-well plates, incubated at 37°C/5% $CO_2$ incubator and observed daily for cytotoxicity for up to 7 days. We also tested eNAT and AVL buffers directly and by dilution on Vero E6 cell lines without prior purification to confirm cytotoxicity in the absence of any purification steps, which showed cytotoxicity until 1:1000 dilution.

## Inactivation of SARS-CoV-2 treated with eNAT

Viral inactivation with eNAT was explored in two different approaches; 1) by spiking the virus directly to the eNAT, simulating a swab sample that is placed into a transport media; and 2) by diluting a clinical sample such as saliva in eNAT, simulating a self-collected sample transport scenario. For the first approach, 100μl of the SARS-CoV-2 virus culture ($8\times10^6$ PFU/ml) was added to 2ml of eNAT, mixed and incubated for 0, 1, 2, 5, 10 and 15 min at room temperature. After incubation, the entire sample was processed using Pierce 4ml DRSC to remove cytotoxic components from eNAT. For the second approach, one-ml of confirmed SARS-CoV-2 negative saliva was spiked with 100 μl of SARS-CoV-2 ($3\times10^6$ PFU/ml) and then added to eNAT at 1:1 and 1:2 sample to eNAT ratio. The samples were incubated at room temperature for 10, 15 and 30 min for the 1:1 ratio and 5, 10, 15 min for the 1:2 ratio. Samples were then processed with the DRSC columns as described above. We performed two negative controls for both approaches. The first negative control consisted of 100 μl of virus free DMEM, which was added to 2 ml of eNAT at both 1:1 and 1:2 ratio; the second negative control consisted of 100 μl of the SARS-CoV-2 virus that was spiked into 400 μl of AVL buffer (Qiagen, Germantown, MD) and then heated to 92˚C for 15 min to inactivate the virus. As a positive control, 100 μl of SARS-CoV-2 was spiked into 2 ml of DMEM. Positive control samples were also assessed before and after DRSC purification to determine viral loss during this process. All controls were performed in either duplicate or triplicate for each experiment.

## Calculating eNAT sterilizing activity

Viral cytopathic effect (CPE) was determined in all samples by both direct infection in Vero E6 cells in 6-well plates and TCID50 assay by serial diluting of the samples on 96-well plates. For TCID50 assay, samples were diluted ten-fold in DMEM+2%FBS and titers were calculated by Reed & Muench method [17]. Viral titer reduction was determined by subtracting the viral titer for the treated samples from the untreated samples. For serial passaging of the samples, each 2 ml replicate of the DRSC column purified sample was added into 4 different wells (500 μl of the sample in each well) in a 6-well plate and incubated at 37˚C/5% $CO_2$ for up to 14 days, with two passages every 5 to 7 days. The plates were checked daily under an inverted microscope for the presence of CPE. All samples were also tested by RT-PCR at days 7 and 14 of serial passaging using the Xpert Xpress SARS-CoV-2 test (Xpert, Cepheid, Sunnyvale, CA). RT-PCR Ct values were expected to decrease after multiple passages on Vero E6 cells if replicative virions were present, while RT-PCR Ct values were expected to be very delayed or absent if few or no virions were present. The Xpert test reports amplification of sequences of the envelope (E) and nucleocapsid (N2) genes. Positive results were indicated by the detection of either the N2 target or both E and N2 targets.

## Integrity of SARS-CoV-2 RNA in eNAT

The stability of SARS-CoV-2 RNA in eNAT was evaluated for a period of 7 days directly by RT-PCR. Negative saliva samples were spiked with SARS-CoV-2 virus at 5X the LOD (0.05 PFU/ml) previously established for nasopharyngeal (NP) matrix using the Xpert test [18]. Samples were either swabbed and/or diluted 1:2 in eNAT. All samples were stored at three different temperatures (4˚C, 28˚C and 35˚C) and viral RNA was measured at different time points (0h, 4h, 24h, 48h and 168 h) using the Xpert test.

**Table 1. Effectiveness of different methods at removing basal cytotoxicity from eNAT transport media.**

| Reagent | Methods | Total processing time | Cytotoxicity in VeroE6 cell line | Percentage of virus recovery |
|---------|---------|-----------------------|----------------------------------|------------------------------|
| eNAT | Slide-A-Lyzer G2 dialysis cassette (10K MWCO) | Overnight (16-18h) | No | 100% |
| | Pierce 4ml DRSC | 12 min | No | 100% |
| | Amicon Ultra 4ml (100 KDa) | 40 min | Yes | Not tested |
| | PD10 desalting column | 15 min | Yes | Not tested |

## Statistical analysis

Microsoft Excel office 365 and GraphPad Prism 8.4.3 for Windows were used for graphing and analysis of the data. Ordinary one-way ANOVA with Tukey multiple comparisons was applied to calculate P-values.

## Results

### Removal of basal cytotoxicity

Both eNAT and AVL buffers are cytotoxic to Vero E6 cells when used directly. We explored four different methods of purification as listed in Table 1. Microscopic observation of the Vero E6 cells in a 6-well plate revealed that the buffers purified through DRSC columns and Slide-A-Lyzer G2 dialysis cassettes did not show any cytotoxic effect to Vero E6 cells for up to 7 days. However, the filtrates from PD 10 and Amicon ultra columns were cytotoxic within 24 hours (Table 1).

### Inactivation of SARS-CoV-2 with eNAT

eNAT, a guanidine-thiocyanate based sample transport medium is claimed to maintain the integrity of nucleic acids for long periods (https://www.copanusa.com/sample-collection-transport-processing/enat/). We evaluated the ability of eNAT as a transport media to inactivate SARS-CoV-2 directly (simulating swab samples) or when diluted with saliva. The virus was added directly to eNAT at the final concentration of $4\times10^5$ PFU/ml, incubated for 0, 1, 2, 5, 10 and 15 min and filtered using DRSC columns. Vero E6 cells were infected with the flow through from these samples and observed for cytopathic effect for up to 14 days with two passages. All test samples and controls were processed and tested on Vero E6 cells in the same manner. We found that the negative controls did not cause any CPE while a 100 µl positive control containing $4\times10^5$ PFU/ml of SARS-CoV-2 resulted in high CPEs within 48–72 hours of infection in Vero E6 cells. After 14 days of incubation and two passages, both replicates of the test samples incubated in eNAT for 0 min produced CPEs and were positive for SARS-CoV-2 by RT-PCR with an average N2 cycle threshold (Ct) 37.4±1.6 (Table 2). We did not observe any visible CPE after 1 min of incubation in eNAT; however, the RT-PCR assay was positive in 1 out of 2 replicates (Ct 37.4). Extending eNAT incubation to 2 min through 15 mins eliminated both CPE and any RT-PCR positivity, demonstrating a $>5.6 \log_{10}$ PFU/ml reduction in the viral load (Table 2).

We further explored the ability of eNAT to inactivate SARS-CoV-2 when diluted in a sample such as saliva. We spiked SARS-CoV-2 ($3\times10^6$ PFU/ml) into SARS-CoV-2 negative saliva and then incubated the samples with eNAT at 1:1 and 1:2 (sample to eNAT) ratios for different time points at room temperature. After 14 days of incubation and 2 passages, eNAT alone and AVL buffer negative controls, maintained cell integrity without any visible CPEs and positive control showed CPEs within 48 hours (N2-Ct = 13.8; Table 3). Although we did not observe any CPEs under any test conditions, the 1:1 saliva/eNAT mixtures were found to be SARS-

**Table 2. Inactivation of SARS-CoV-2 treated directly with eNAT transport media.**

| Reagents | Sample matrix | Incubation time (minutes) | Presence of CPE[a] (replicates) | Presence of viral RNA[b] (replicates; Ct±SD) |
|---|---|---|---|---|
| eNAT | Virus in tissue culture media | 0 | Yes (2/2) | Yes (2/2; 37.4±1.6) |
| | | 1 | No (0/2) | Yes (1/2; 37.4±0) |
| | | 2 | No (0/2) | No (0/2) |
| | | 5 | No (0/2) | No (0/2) |
| | | 10 | No (0/3) | No (0/3) |
| | | 15 | No (0/3) | No (0/3) |
| | Tissue culture media without the virus | None | No (0/2) | No (0/2) |
| AVL buffer + heating at 92˚C (Negative control) | Virus in tissue culture media | 15 | No (0/2) | No (0/2) |
| None (Positive control) | Virus in tissue culture media | None | Yes | Yes (13.6) |

[a]Tested by both CPE and TCID50;
[b]Tested by Xpert Xpress SARS-CoV-2 test.

CoV-2 positive by RT-PCR with N2-Ct >40 in all replicates (3/3) after 10 and 15 min of incubation. Even after 30 min of incubation in eNAT, 2/3 samples were positive for SARS-CoV-2 by RT-PCR. However, the spiked saliva samples that were mixed with 1:2 ratios of eNAT appeared to be completely inactivated (calculated SARS-CoV-2 inactivation efficacy >5.2 $\log_{10}$ PFU/ml) after as little as 5 min incubation with the eNAT.

## Stability of SARS-CoV-2 RNA in eNAT

We evaluated the stability of the SARS-COV-2 RNA in saliva treated with eNAT over a range of storage times and temperatures. SARS-COV-2 virus was spiked into saliva samples at 5X the established NP limit of detection (LOD, 0.05 PFU/ml) of the Xpert test [18], and the samples were added to eNAT either as swab samples or saliva/eNAT mixed at 1:2 ratio. The samples were stored at 4˚C, 28˚C and 35˚C and were periodically tested for the presence of SARS-CoV-2 by RT-PCR after 0, 4, 24, 48 and 168 h (7 days) of storage. As shown in Fig 1A, the

**Table 3. Inactivation of SARS-CoV-2 in saliva samples diluted with eNAT.**

| Reagent | Sample matrix | Sample to eNAT ratio | Incubation time (min) | Presence of CPE[a] (replicates) | Presence of viral RNA[b] (replicates; Ct±SD) |
|---|---|---|---|---|---|
| eNAT | Saliva with virus | 1:1 | 10 | No (0/3) | Yes (3/3; 41.2±1.1) |
| | | | 15 | No (0/3) | Yes (3/3; 41.1±1.0) |
| | | | 30 | No (0/3) | Yes (2/3; 42.3±1.9) |
| | | 1:2 | 5 | No (0/3) | No (0/3) |
| | | | 10 | No (0/3) | No (0/3) |
| | | | 15 | No (0/3) | No (0/3) |
| | Tissue culture media without the virus | 1:1 | None | No (0/2) | No (0/2) |
| | | 1:2 | None | No (0/2) | No (0/2) |
| AVL buffer + heating at 92˚C (Negative control) | Virus in tissue culture media | None | 15 | No (0/2) | No (0/2) |
| None (Positive control) | Virus in tissue culture media | None | None | Yes | Yes (13.8) |

[a]Tested by both CPE and TCID50;
[b]Tested by Xpert Xpress SARS-CoV-2 test.

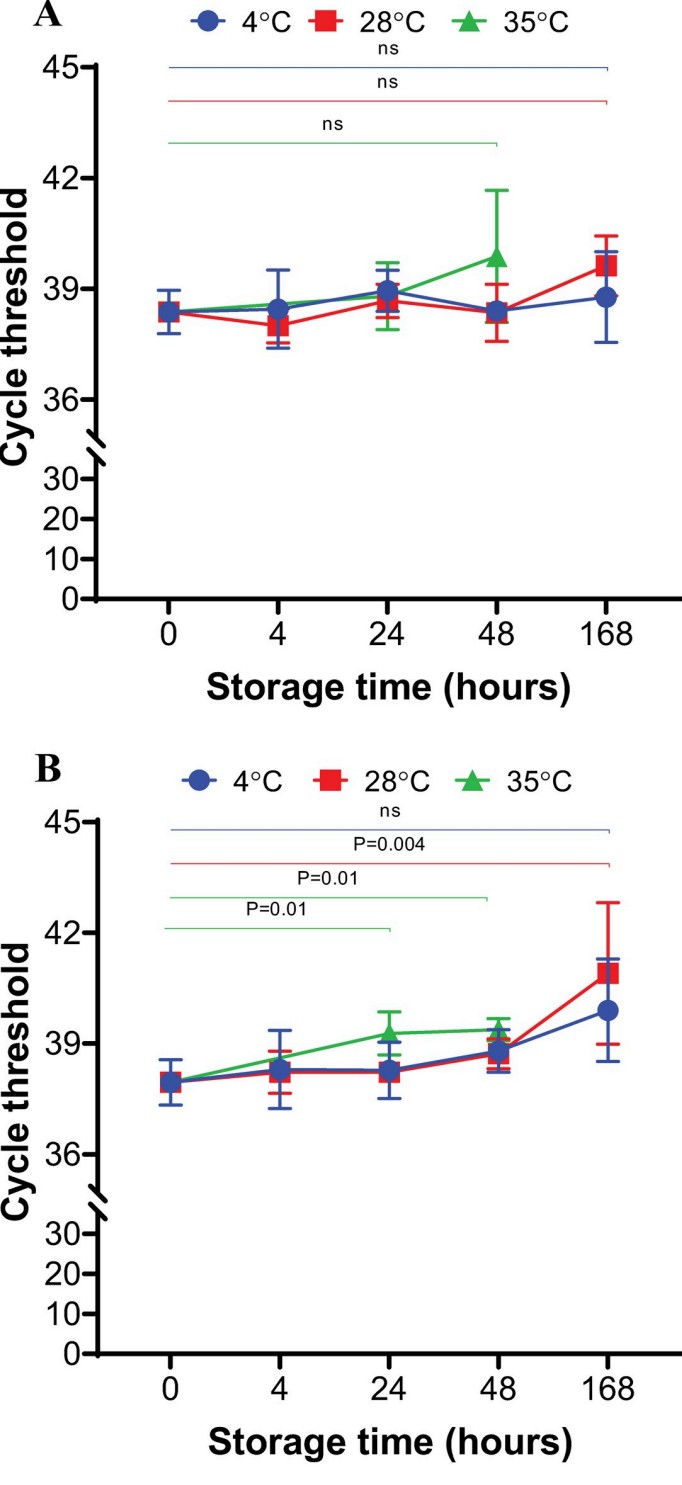

**Fig 1. Stability of SARS-CoV-2 RNA in a swab of saliva mixed with eNAT (A) or saliva diluted 1:2 with eNAT (B).** Saliva samples were spiked with SARS-CoV-2 virus at 5X the LOD (0.05 PFU/ml) and maintained at 4˚C, 28˚C (room temperature) and 35˚C. The samples were tested for SARS-CoV-2 RNA by RT-PCR at 0, 4, 24, 48 and 168 hours. The samples kept at 35˚C were tested at 0, 24 and 48 hours. N2 gene cycle threshold (Ct) values are shown. Four replicates were performed for each condition and results are expressed as mean±SD. ns-not statistically significant. P<0.05 is considered statistically significant.

RNA was stable for the swab samples stored in eNAT under all conditions as indicated by N2 gene cycle threshold (Ct) (P>0.05). In saliva samples diluted 1:2 in eNAT, the RNA was stable until 168 h at 4˚C and for 48 h at 28˚C (P>0.05, Fig 1B). However, average N2-Ct increased by 3 Ct values at 28˚C after 7 days (P = 0.004), as well as a significant decrease in RNA in as little as 24 hours in samples incubated at 35˚C (P = 0.01). The RNA detection rate remained at 100% positive rate for all samples with very low viral load, through all tested conditions despite the increase in assay Ct values over time and increased temperature.

## Discussion

We have demonstrated that eNAT can rapidly inactivate SARS-CoV-2 present in saliva samples when used at a ratio of 1:2 (sample to eNAT) or higher. Furthermore, SARS-CoV-2 RNA is stable in eNAT for at least 48 hours at room temperature or below even when tested at very low viral load (0.05 PFU/ml). These results strongly suggest that eNAT can be used to ensure safe handling, collection, transport, storage, and processing of specimens intended to be tested for SARS-CoV-2 using the Xpert test. The safety provided using a viral inactivating buffer may simplify testing in several settings, including home sample collection and rapid assay testing outside of conventional laboratories. Recent studies have shown that buffers containing guanidinium thiocyanate can inactivate SARS-CoV-2 [5,7,9,12,13,19,20]. In this study, we conclusively demonstrated that eNAT, a guanidinium thiocyanate-based buffer can be used as a viral inactivation and stabilization media for RT-PCR based detection of SARS-CoV-2 in clinical specimens.

Saliva is a popular alternative specimen for COVID testing because of its ease of self-collection. However, samples transported by mail can leak due to improper packaging or handling, raising safety concerns [4]. Our studies demonstrate that eNAT can inactivate SARS-CoV-2 spiked saliva samples within 5 minutes when used in 2 volumes of eNAT to 1 volume of the sample. To our knowledge, this is the first study evaluating eNAT for inactivation of saliva samples spiked with high titer of SARS-CoV-2. These results can help guide practices for safe specimen handling, transport, and processing, and may permit safe SARS-CoV-2 testing in locations without strict BSL2 practices and/or decontamination procedures.

To accurately evaluate the efficacy of viral inactivation, we established rapid methods to remove any background cytotoxicity caused by the buffer in the absence of the virus. We explored several methods involving dilution, dialysis, and column filtration [7,21–23] Both dialysis and filtration using Pierce DRSC spin columns resulted in complete removal of basal cytotoxicity from eNAT and did not result in any viral loss (Table 1). However, our preference was to use DRSC spin columns because this method detoxified samples within minutes compared to the many hours required by a dialysis process. The more rapid processing also ensured that we were able to make accurate determinations of buffer inactivation time. Dilution methods might have been an alternate way to remove cytotoxic components; however, we have observed that eNAT remained cytotoxic even after a 1000-fold dilution when tested directly on the cells and such high dilutions can produce misleading estimates of log titer virus inactivation unless very high titer viral stocks can be tested [7,13].

We suggest that eNAT can be used as part of SARS-CoV-2 RT-PCR testing programs in schools, workplaces, prisons, skilled nursing facilities, homeless shelters, and other high-risk locations, perhaps in combination with sample pooling strategies to decrease total testing costs. The use of eNAT is further supported by our determination that eNAT can help increase the clinical sensitivity of the Xpert Xpress SARS-CoV-2 test when used with variety of upper respiratory specimens [24]. While the use of a sterilizing buffer is only one component of a

successful worldwide testing program, the utility of this type of reagent and its potential contributions to designing improved testing strategies should be explored.

## Acknowledgments

We thank the World Reference Center for Emerging Viruses and Arboviruses, the University of Texas Medical Branch, Galveston, Tx, for providing the SARS-CoV-2 USA-WA1/2020 strain.

## Author Contributions

**Conceptualization:** Sukalyani Banik, Glenn Johns, Robert Kwiatkowski, David Alland, Padmapriya P. Banada.

**Data curation:** Sukalyani Banik, Padmapriya P. Banada.

**Formal analysis:** Sukalyani Banik, Padmapriya P. Banada.

**Funding acquisition:** David Alland.

**Investigation:** Sukalyani Banik, Kaheerman Saibire, David Alland, Padmapriya P. Banada.

**Methodology:** Sukalyani Banik, Kaheerman Saibire, Shraddha Suryavanshi, Robert Kwiatkowski, David Alland, Padmapriya P. Banada.

**Project administration:** Glenn Johns, Robert Kwiatkowski, David Alland.

**Resources:** Glenn Johns, Soumitesh Chakravorty, Robert Kwiatkowski.

**Supervision:** Glenn Johns, Robert Kwiatkowski, David Alland.

**Writing – original draft:** Sukalyani Banik, Padmapriya P. Banada.

**Writing – review & editing:** Sukalyani Banik, Glenn Johns, Soumitesh Chakravorty, Robert Kwiatkowski, David Alland, Padmapriya P. Banada.

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
