## [Decision Letter · Decision Letter 0]

12 Apr 2021

PONE-D-21-08255

Inactivation of SARS-CoV-2 virus in saliva using a guanidium based transport medium suitable for RT-PCR diagnostic assays.

PLOS ONE

Dear Dr. Banada,

Thank you for submitting your manuscript to PLOS ONE. After careful consideration, we feel that it has merit but does not fully meet PLOS ONE’s publication criteria as it currently stands. Therefore, we invite you to submit a revised version of the manuscript that addresses the points raised during the review process.

We look forward to receiving your revised manuscript.

Kind regards,

Ruslan Kalendar, PhD

Academic Editor

PLOS ONE

Journal Requirements:

3) Please provide additional details regarding participant consent. In the ethics statement in the Methods and online submission information, please ensure that you have specified (i) whether consent was informed and (ii) what type you obtained (for instance, written or verbal, and if verbal, how it was documented and witnessed). If your study included minors, state whether you obtained consent from parents or guardians. If the need for consent was waived by the ethics committee, please include this information.

4)  Thank you for stating the following in the Competing Interests section:

'R.K. G.J, and S.C. are employees of Cepheid Inc., which sells the Xpert Xpress SARSCoV-2 test. D.A. receives research support and royalty payments from Cepheid'

5) We note that you have included the phrase “data not shown” in your manuscript. Unfortunately, this does not meet our data sharing requirements. PLOS does not permit references to inaccessible data. We require that authors provide all relevant data within the paper, Supporting Information files, or in an acceptable, public repository. Please add a citation to support this phrase or upload the data that corresponds with these findings to a stable repository (such as Figshare or Dryad) and provide and URLs, DOIs, or accession numbers that may be used to access these data. Or, if the data are not a core part of the research being presented in your study, we ask that you remove the phrase that refers to these data.

Reviewers' comments:

Reviewer's Responses to Questions

**Comments to the Author**

1. Is the manuscript technically sound, and do the data support the conclusions?

Reviewer #1: Yes

Reviewer #2: Yes

Reviewer #3: Yes

Reviewer #4: Partly

2. Has the statistical analysis been performed appropriately and rigorously? 

Reviewer #1: N/A

Reviewer #2: Yes

Reviewer #3: Yes

Reviewer #4: Yes

3. Have the authors made all data underlying the findings in their manuscript fully available?

Reviewer #1: Yes

Reviewer #2: No

Reviewer #3: Yes

Reviewer #4: No

4. Is the manuscript presented in an intelligible fashion and written in standard English?

Reviewer #1: Yes

Reviewer #2: Yes

Reviewer #3: Yes

Reviewer #4: Yes

5. Review Comments to the Author

 Reviewer #1: 

Banik et. al. describes the use of a commercially available buffer to inactivate SARS-CoV-2 virus in saliva sample which is compatible with downstream RT-PCR applications. The research question is clear, and the methods used and results are adequate to answer the research question. The paper is well written in general. There are some minor comments

Reviewer #2: 

This manuscript demonstrates eNAT transport buffer inactivates infectious SARS-CoV-2 virus in contrived swab and saliva samples. The authors also quantify SARS-CoV-2 RNA stability in eNAT for various lengths of time and at different temperatures. Overall the manuscript is technically sound and the data support the conclusions. However, not all data described is presented by the authors — the manuscript will be ready for publication following clarification of the following points:

* Lines 116-119 and 251-252 refer to data not shown in this manuscript, which is not permitted by PLoS journals. Please remove this text or include the referenced data.

* Lines 123-124 of the Methods state: “For the first approach, 100µl of the SARS-CoV-2 virus culture (8X106 PFU/ml) was added to 2ml of eNAT”. However, lines 174-174 of the Results state: “The virus was added directly to eNAT at the final concentration of 8X105 PFU/ml”. If 100ul of 8e6 PFU/mL was added to 2mL, the final concentration would be ~4e5 PFU/mL. Please clarify.

* Lines 185-186 describe the “>5.6 log10 PFU/ml reduction in viral load” following mock swab inactivation with eNAT. 5.6 log10 PFU/ml is roughly equivalent to 4e5 PFU/mL so this statement is accurate if this was the viral treatment titer (see previous point). However, line 199 repeats that the inactivation efficiency is >5.6 log10 PFU/mL for virus-spiked saliva, even though the authors used a less concentrated viral stock to generate the contrived saliva samples (100ul 3e6 PFU/mL into 1ml saliva = ~3e5 PFU/ml). How can the inactivation efficiency be the same for mock swab and saliva samples when the titer of the mock saliva samples was lower? Please update the statement about saliva sample inactivation (or elaborate on how these calculations were performed).

* Lines 245-247 state: “Both dialysis and filtration using Pierce DRSC spin columns resulted in complete removal of basal cytotoxicity from eNAT and did not result in any viral loss”. Additionally, lines 135-137 state: “Positive control samples were also assessed before and after DRSC purification to determine viral loss during this process.” However, this manuscript does not present data demonstrating viral titers are equivalent pre and post dialysis/filtration. Please amend the claim in 245-247 and remove lines 135-137 (or include equivalency data supporting this claim).

Reviewer #3: 

The manuscript is technically sound, with appropriate controls. It is well written and clear, easy to understand. The results have useful implicatios for sample testing as explained in the intro/discussion.

A few minor points:

1. In intro, lines 38-39: suggest define sars-cov-2 and covid19 upon first use.

2. lines 135-137 explain that a positive control was included to show no loss from DRSC (which is excellent, sometimes this is overlooked so great to include), can these data be added/explained to show there's no loss? (currently it seems results just say there was high CPE, but an actual titer comparison would be useful, as if there is some loss of infectivity from DRSC, then a lower concentration of virus [e.g. if only partial inactivation happened] would look completely negative due to the DRSC, rather than due to the eNAT).

3. method lines 148-150 explain the RT-PCR assays, but can it be more clearly stated in results what is being measured in each section (e.g. "live" infectious virus in the inactivation sections, vs. just RNA copies later on?). The PCR results are used to show inactivation, for example in Table 2, so negative on PCR means virus is inactivated, but one would expect RNA to still be present (which is shown later in stability section), so different things are being measured via RT-PCR in the different sections. But if this can be briefly, clearly described that might be beneficial for some readers.

Reviewer #4: 

Review Summary

The manuscript titled, “Inactivation of SARS-CoV-2 virus in saliva using a guanidium based transport medium suitable for RT-PCR diagnostic assays” is concise and well-written. It clearly demonstrates the effectiveness of eNAT, a guanidium thiocyanate containing buffer, at rapidly inactivating live SARS-CoV-2 virus when used at a ratio of 2:1 with saliva or as a transport medium for swab specimen. eNAT supported detection of low concentrations of SARS-CoV-2 RNA for at least 7 days at a range of storage temperatures with almost no loss of sensitivity. The manuscript suggests in passing that the eNAT buffer also preserves the specimen, although this is not explicitly addressed by the data. There are some minor inconsistencies in the reported concentrations of SARS-CoV-2 virus used in the manuscript, but they do not affect the conclusions of the paper. Overall, the research presented here is of use to the public health and clinical testing communities and represents a useful addition to the available arsenal of specimen collection mechanisms, but the scope of the findings is necessarily narrow.

1. Is the manuscript technically sound, and do the data support the conclusions?

Partly: The manuscript is technically sound, but the reported viral concentrations and dilutions don’t totally add up. The data support the main conclusion that eNAT can inactivate SARS-CoV-2 in saliva at a ratio of 1:2 (Saliva:eNat) in as little as 5 minutes.

2. Has the statistical analysis been performed appropriately and rigorously?

Yes.

3. Have the authors made all data underlying the findings in their manuscript fully available?

Partly: The data from the CPE and TCID50 dilution series are only presented in a final summary. I don’t know if it is customary to present the entirety of this type of data, but I have seen it presented in other manuscripts.

4. Is the manuscript presented in an intelligible fashion and written in standard English?

Yes, the manuscript is well written and easy to read.

Important Points

1. In the ”Inactivation of SARS-CoV-2 with eNAT” results section it states that SARS-CoV-2 is diluted to a final concentration of 8x10^5 PFU/mL in eNAT. The methods section describes diluting 100uL of 8x10^6 PFU/mL in 2mL of eNAT which should give you a final concentration of ~4x10^5 PFU/mL.

a. The results describe a “100uL positive control containing 8x10^6 PFU/mL of SARS-CoV-2” but the methods suggest that this control was further diluted in 2mL of DMEM to the same concentration as the eNAT test samples (~4x10^5 PFU/mL).

b. Ideally the concentrations of virus should be reported consistently throughout the manuscript as either the final diluted concentration or the stock concentration into a known dilution volume.

c. The 8x10^5 PFU/mL diluted concentration does not make sense given the stock concentration and dilution volumes listed

2. Although the SARS-CoV-2 titers used in the manuscript are reasonably high, it is not uncommon to see quantities of genomic fragments in saliva on the order of 10^8-10^10 copies/mL (~10% of cases) and >10^10 copies/mL in a smaller number of cases (~2%) based on the preprint below, using qRT-PCR. I see similar results using ddPCR for quantification. Although these PCR-based methods do not quantify infectious virions, is an inactivation efficacy >5.6 log10 PFU/ml an appropriate benchmark for SARS-CoV-2 inactivation of saliva samples?

a. Yang Q, Saldi TK, Lasda E, Decker CJ, Paige CL, Muhlrad D, Gonzales PK, Fink MR, Tat KL, Hager CR, Davis JC, Ozeroff CD, Meyerson NR, Clark SK, Fattor WT, Gilchrist AR, Barbachano-Guerrero A, Worden-Sapper ER, Wu SS, Brisson GR, McQueen MB, Dowell RD, Leinwand L, Parker R, Sawyer SL. Just 2% of SARS-CoV-2-positive individuals carry 90% of the virus circulating in communities. medRxiv [Preprint]. 2021 Mar 5:2021.03.01.21252250. doi: 10.1101/2021.03.01.21252250. PMID: 33688663; PMCID: PMC7941634.

3. The evaluation of stability of SARS-CoV-2 RNA in eNAT shows that eNAT does not interfere with qRT-PCR detection of viral RNA over a course of 7 days, but there is no data showing that eNAT improves the stability of viral RNA compared to a similar specimen without an inactivating substance. My experience is that raw saliva is stable for at least 7 days after heat inactivation without the need for preservatives, and this is documented in the SalivaDirect protocol.

a. In the discussion the authors conclude, “eNAT, a guanidinium thiocyanate-based buffer can be used as a viral inactivation and preservation media for SARS-CoV-2 specimens.”

b. I don’t think the data show that eNAT “preserves” the specimen. This may be a matter of nomenclature, where any media that doesn’t degrade the sample is considered to preserve the sample, but in my mind a preservative prevents normal degradation, and there is no evidence in this manuscript that eNAT reduces the normal degradation of RNA in saliva.

c. My experience with heat-inactivated saliva is that the RNA is severely degraded but still detectable by RT-PCR. Any data or citation showing a reduction in degradation of RNA in saliva mixed with eNAT would be strong evidence that eNAT does stabilize/preserve the specimen.

Minor Points

1. From a practical standpoint, it is difficult to see how eNAT would be implemented in a saliva collection aid, which would need to come pre-aliquoted with a generous amount of eNAT to account for the variable amount of saliva collected. This does not affect the findings of the manuscript, but I am curious how the authors envision the use of this product.

2. There is only a brief description of the Xpert Xpress SARS-CoV-2 test. Since this test was designed and validated for upper respiratory specimen such as nasopharyngeal swabs but is being applied to saliva specimen perhaps a more detailed explanation is appropriate. For instance, from what I could gather the volume of sample loaded into the Xpert Xpress cartridge is ~300uL. This is a significant deviation from standard RT-PCR protocols which use on the order of 20uL total for a single reaction.

a. The authors only report results for the N2 target gene in the manuscript. This is certainly adequate to monitor for the presence of viral RNA, but is there a reason the results for the E target were omitted? Was there any interference from the eNAT on the amplification of the E target?

6. PLOS authors have the option to publish the peer review history of their article (what does this mean?). If published, this will include your full peer review and any attached files.

Reviewer #1: No

Reviewer #2: No

Reviewer #3: No

Reviewer #4: **Yes: **Philip D Fox

---

## [Author Response · Author response to Decision Letter 0]

18 May 2021

Reviewer #1: 

Banik et. al. describes the use of a commercially available buffer to inactivate SARS-CoV-2 virus in saliva sample which is compatible with downstream RT-PCR applications. The research question is clear, and the methods used and results are adequate to answer the research question. The paper is well written in general. There are some minor comments 

Banik et. al. describes the use of a commercially available buffer to inactivate SARS-CoV-2

virus in saliva sample which is compatible with downstream RT-PCR applications. The

research question is clear, and the methods used and results are adequate to answer the

research question. The paper is well written in general.

Question:

Q: How common is saliva swab/sample being used for SARS-CoV-2 detection compared to

nasal swab? Have you tested eNAT inactivation on nasal swab which is commonly used?

Ans: Saliva is a commonly used and popular non-invasive sample type for detection of SARS-CoV-2 from patients. In another related study, which is currently under review in a different journal and published in the preprint journal MedRXIV (Banada et al., https://doi.org/10.1101/2021.03.03.21251172), we demonstrated that a saliva swab in eNAT buffer harbors more virions and thus provides a higher detection rate compared to a nasal swab. In this study, we have demonstrated that SARS-CoV-2 spiked directly into eNAT transport media, which simulates any swab specimen in the media, completely inactivates SARS-CoV-2 spiked in at a concentration as high as 4x105 PFU/ml, although a nasal swab was specifically not tested.

Minor comments:

Q: Line 20: spell out RT-PCR

Ans: corrected as suggested in the revised manuscript (Line 20-21).

Q: Line 40: instead of “after being opened” it is better to say “during processing”

Ans: corrected as suggested in the revised manuscript (Line 42).

Q: Line 61: instead of “be used”, it is better to use “help”

Ans: corrected as suggested in the revised manuscript (Line 63).

Q: Line 68 and 127 Use “SARS-CoV-2” negative instead of “COVID-19”

Ans: corrected as suggested in the revised manuscript (Lines 70 and 129).

Q: Line 82, 87, 95, 124, 128, 153, 175 etc.: change “X” to “´×”

Ans: corrected as suggested throughout in the revised manuscript, where concentrations are mentioned.

Q: Line 103: Please add the MWCO for the Slide-A-Lyzer

Ans: The 10K MWCO is now added in the revised manuscript (Line 105).

Q: Line 161, 162: change cell “lines” to “line”

Ans: corrected as suggested throughout the revised manuscript where cell line is mentioned.

Q: Table 1: Please add the MWCO for “Amicon Ultra 4mL” and Slide A-Lyzer

Ans: Amicon ultra 100 KDa and 10K MWCO for Slide A-lyzer is now included in the Table 1 of revised manuscript. 

Q:Table 2 and 3: “Tissue” to “tissue”

Ans: corrected as suggested in the revised manuscript where tissue culture media was mentioned (Table 2 and 3).

Q: Table 2 and 3: I suggest centre the text in the table

Ans: corrected as suggested in the revised manuscript (Table 2 and 3).

Q: Table 2 and 3: “tested by both culture and TCID50” to “tested by both CPE and TCID50”

Ans: corrected as suggested in the revised manuscript (Table 2 and 3).

Q: Line 217: define “LOD”

Ans: LOD is the limit of detection of the assay used in this study. A previous publication on the Xpert Xpress SARS-CoV-2 test defined and established the LOD (Loeffelholz, Alland et al. 2020), which was used as a reference, in our study. 

Q: Statistical analysis

What’s the sample size in Figure 1? If the sample size is small and data is not normally

distributed, a non-parametric test (Krusal-Wallis) is more appropriate.

What post hoc test was used on top of ANOVA (or Kruskal-Wallis)? Please provide the information in the manuscript or perform post-hoc test is it hasn’t been done.

Ans: Ordinary One-way ANOVA with Tukey multiple comparison analysis was done using Graphpad Prism. This information is included in the revised manuscript. 

 

Reviewer #2: 

This manuscript demonstrates eNAT transport buffer inactivates infectious SARS-CoV-2 virus in contrived swab and saliva samples. The authors also quantify SARS-CoV-2 RNA stability in eNAT for various lengths of time and at different temperatures. Overall the manuscript is technically sound and the data support the conclusions. However, not all data described is presented by the authors — the manuscript will be ready for publication following clarification of the following points:

Q:* Lines 116-119 and 251-252 refer to data not shown in this manuscript, which is not permitted by PLoS journals. Please remove this text or include the referenced data.

Ans: Data was written as a statement in the methods section (line 121) thus, ‘data not shown’ is now removed from the discussion section of the revised manuscript (Line 259).

Q* Lines 123-124 of the Methods state: “For the first approach, 100µl of the SARS-CoV-2 virus culture (8X106 PFU/ml) was added to 2ml of eNAT”. However, lines 174-174 of the Results state: “The virus was added directly to eNAT at the final concentration of 8X105 PFU/ml”. If 100ul of 8e6 PFU/mL was added to 2mL, the final concentration would be ~4e5 PFU/mL. Please clarify.

Ans: The reviewer is correct, and we have corrected this oversight in the revised manuscript (Lines 181 and 185). We thank the reviewer for catching this error.

Q:* Lines 185-186 describe the “>5.6 log10 PFU/ml reduction in viral load” following mock swab inactivation with eNAT. 5.6 log10 PFU/ml is roughly equivalent to 4e5 PFU/mL so this statement is accurate if this was the viral treatment titer (see previous point). However, line 199 repeats that the inactivation efficiency is >5.6 log10 PFU/mL for virus-spiked saliva, even though the authors used a less concentrated viral stock to generate the contrived saliva samples (100ul 3e6 PFU/mL into 1ml saliva = ~3e5 PFU/ml). How can the inactivation efficiency be the same for mock swab and saliva samples when the titer of the mock saliva samples was lower? Please update the statement about saliva sample inactivation (or elaborate on how these calculations were performed).

Ans: The statement has been updated in the revised manuscript with the accurate numbers (Line 205).

Q:* Lines 245-247 state: “Both dialysis and filtration using Pierce DRSC spin columns resulted in complete removal of basal cytotoxicity from eNAT and did not result in any viral loss”. Additionally, lines 135-137 state: “Positive control samples were also assessed before and after DRSC purification to determine viral loss during this process.” However, this manuscript does not present data demonstrating viral titers are equivalent pre and post dialysis/filtration. Please amend the claim in 245-247 and remove lines 135-137 (or include equivalency data supporting this claim).

Ans: The statement regarding the positive control is accurate and is presented in the revised Table 1 of the revised manuscript.

 

Reviewer #3: 

The manuscript is technically sound, with appropriate controls. It is well written and clear, easy to understand. The results have useful implicatios for sample testing as explained in the intro/discussion.

A few minor points:

Q: 1. In intro, lines 38-39: suggest define sars-cov-2 and covid19 upon first use.

Ans: SARS-CoV-2 and COVID-19 are now defined briefly in the introduction of the revised manuscript (Lines 37-40).

Q:2. lines 135-137 explain that a positive control was included to show no loss from DRSC (which is excellent, sometimes this is overlooked so great to include), can these data be added/explained to show there's no loss? (currently it seems results just say there was high CPE, but an actual titer comparison would be useful, as if there is some loss of infectivity from DRSC, then a lower concentration of virus [e.g. if only partial inactivation happened] would look completely negative due to the DRSC, rather than due to the eNAT).

Ans: The data has now been added to Table 1 of the revised manuscript.

Q: 3. method lines 148-150 explain the RT-PCR assays, but can it be more clearly stated in results what is being measured in each section (e.g. "live" infectious virus in the inactivation sections, vs. just RNA copies later on?). The PCR results are used to show inactivation, for example in Table 2, so negative on PCR means virus is inactivated, but one would expect RNA to still be present (which is shown later in stability section), so different things are being measured via RT-PCR in the different sections. But if this can be briefly, clearly described that might be beneficial for some readers.

Ans: As correctly stated by the reviewer, yes in this experiment we have used RT-PCR as a measure for viral growth to study inactivation. After two passages on Vero E6 cells, viral RNA is not expected to be present since there are no replicative viruses present, if the virus is inactivated. If there are very few replicative virions present, then viral RNA will be detected at very late Ct values, indicating either incomplete inactivation, which was what we observed in our study for initial time points (0 min and 1 min). In contrast, our stability study examined virus that was spiked into the eNAT buffer and was tested by RT-PCR directly, without viral propagation. Here, one would expect to see unchanged Ct values at different conditions and time if the viral stability is preserved by the eNAT buffer. We have included part of this explanation in the revised manuscript to make it clear (Lines 150-152).

 

Reviewer #4: 

Review Summary

The manuscript titled, “Inactivation of SARS-CoV-2 virus in saliva using a guanidium based transport medium suitable for RT-PCR diagnostic assays” is concise and well-written. It clearly demonstrates the effectiveness of eNAT, a guanidium thiocyanate containing buffer, at rapidly inactivating live SARS-CoV-2 virus when used at a ratio of 2:1 with saliva or as a transport medium for swab specimen. eNAT supported detection of low concentrations of SARS-CoV-2 RNA for at least 7 days at a range of storage temperatures with almost no loss of sensitivity. 

Q: The manuscript suggests in passing that the eNAT buffer also preserves the specimen, although this is not explicitly addressed by the data. 

Ans: We have revised the phrase to include stability and not preservation. The stability of the RNA in eNAT is demonstrated conclusively.

Q: There are some minor inconsistencies in the reported concentrations of SARS-CoV-2 virus used in the manuscript, but they do not affect the conclusions of the paper. Overall, the research presented here is of use to the public health and clinical testing communities and represents a useful addition to the available arsenal of specimen collection mechanisms, but the scope of the findings is necessarily narrow.

Q1. Is the manuscript technically sound, and do the data support the conclusions?

Partly: The manuscript is technically sound, but the reported viral concentrations and dilutions don’t totally add up. The data support the main conclusion that eNAT can inactivate SARS-CoV-2 in saliva at a ratio of 1:2 (Saliva:eNat) in as little as 5 minutes.

Ans: we agree with the reviewer there was a mistake with the concentrations. But we have corrected that in the revised manuscript. 

Q2. Has the statistical analysis been performed appropriately and rigorously?

Yes.

Q3. Have the authors made all data underlying the findings in their manuscript fully available?

Partly: The data from the CPE and TCID50 dilution series are only presented in a final summary. I don’t know if it is customary to present the entirety of this type of data, but I have seen it presented in other manuscripts.

Q4. Is the manuscript presented in an intelligible fashion and written in standard English?

Yes, the manuscript is well written and easy to read.

Ans: none needed to Q2-Q4. We thank the reviewer for his/her supportive comments.

Important Points

Q: 1. In the ”Inactivation of SARS-CoV-2 with eNAT” results section it states that SARS-CoV-2 is diluted to a final concentration of 8x10^5 PFU/mL in eNAT. The methods section describes diluting 100uL of 8x10^6 PFU/mL in 2mL of eNAT which should give you a final concentration of ~4x10^5 PFU/mL.

a. The results describe a “100uL positive control containing 8x10^6 PFU/mL of SARS-CoV-2” but the methods suggest that this control was further diluted in 2mL of DMEM to the same concentration as the eNAT test samples (~4x10^5 PFU/mL).

Ans: The reviewer is correct, and we have corrected this oversight in the revised manuscript (Lines 181 and 185).

Q: b. Ideally the concentrations of virus should be reported consistently throughout the manuscript as either the final diluted concentration or the stock concentration into a known dilution volume.

Ans: Corrected to maintain consistency throughout the manuscript in the revised copy. 

Q: c. The 8x10^5 PFU/mL diluted concentration does not make sense given the stock concentration and dilution volumes listed

Ans: we have corrected this oversight in the revised manuscript (Line 181).

Q: 2. Although the SARS-CoV-2 titers used in the manuscript are reasonably high, it is not uncommon to see quantities of genomic fragments in saliva on the order of 10^8-10^10 copies/mL (~10% of cases) and >10^10 copies/mL in a smaller number of cases (~2%) based on the preprint below, using qRT-PCR. I see similar results using ddPCR for quantification. Although these PCR-based methods do not quantify infectious virions, is an inactivation efficacy >5.6 log10 PFU/ml an appropriate benchmark for SARS-CoV-2 inactivation of saliva samples?

a. Yang Q, Saldi TK, Lasda E, Decker CJ, Paige CL, Muhlrad D, Gonzales PK, Fink MR, Tat KL, Hager CR, Davis JC, Ozeroff CD, Meyerson NR, Clark SK, Fattor WT, Gilchrist AR, Barbachano-Guerrero A, Worden-Sapper ER, Wu SS, Brisson GR, McQueen MB, Dowell RD, Leinwand L, Parker R, Sawyer SL. Just 2% of SARS-CoV-2-positive individuals carry 90% of the virus circulating in communities. medRxiv [Preprint]. 2021 Mar 5:2021.03.01.21252250. doi: 10.1101/2021.03.01.21252250. PMID: 33688663; PMCID: PMC7941634.

Ans: We agree with the reviewer that a study with even higher concentrations of virus would have been ideal. However, the virus stock used in our manuscript represented that obtained from optimized culture methods followed by 10X concentration. With stocks as high as 8x10^6 PFU/ml, our study is really at or close to the upper limit of virus concentrations that are available. We have spoken to our colleagues who have similar observations. Line 259 does state that studies with even higher concentrations would have been preferable.

Q: 3. The evaluation of stability of SARS-CoV-2 RNA in eNAT shows that eNAT does not interfere with qRT-PCR detection of viral RNA over a course of 7 days, but there is no data showing that eNAT improves the stability of viral RNA compared to a similar specimen without an inactivating substance. My experience is that raw saliva is stable for at least 7 days after heat inactivation without the need for preservatives, and this is documented in the SalivaDirect protocol.

Ans: We agree with the reviewer that the virus in saliva in the absence of any preservatives is stable for at least 7 days, but not inactivated. Here we demonstrated that eNAT can be used both as an inactivating and stabilization buffer for saliva samples from SARS-CoV-2 infected patients even with low viral load. 

a. In the discussion the authors conclude, “eNAT, a guanidinium thiocyanate-based buffer can be used as a viral inactivation and preservation media for SARS-CoV-2 specimens.”

Q: b. I don’t think the data show that eNAT “preserves” the specimen. This may be a matter of nomenclature, where any media that doesn’t degrade the sample is considered to preserve the sample, but in my mind a preservative prevents normal degradation, and there is no evidence in this manuscript that eNAT reduces the normal degradation of RNA in saliva.

Ans: We have revised the phrase to include stability and not preservation (Line 32).

Q: c. My experience with heat-inactivated saliva is that the RNA is severely degraded but still detectable by RT-PCR. Any data or citation showing a reduction in degradation of RNA in saliva mixed with eNAT would be strong evidence that eNAT does stabilize/preserve the specimen.

Ans: To our knowledge, our study is the first to report that eNAT can stabilize and inactivate SARS-CoV-2 RNA in saliva samples. However, since eNAT buffer is a guanidium based buffer, studies have shown that the RNA is stable in guanidium based buffers (Barra, Santa Rita et al. 2021). We have updated the revised manuscript to include these references in line 238.

Minor Points

Q: 1. From a practical standpoint, it is difficult to see how eNAT would be implemented in a saliva collection aid, which would need to come pre-aliquoted with a generous amount of eNAT to account for the variable amount of saliva collected. This does not affect the findings of the manuscript, but I am curious how the authors envision the use of this product.

Ans: eNAT is sold commercially for collection of upper respiratory specimens in 2ml or 3 ml aliquoted tubes with swabs by Copan inc. We used the same in another study where we evaluated different respiratory specimens from patients in different media (published as a preprint in MedRXIV (Banada et al., https://doi.org/10.1101/2021.03.03.21251172). In that study, we added a swab of saliva to eNAT media, which showed comparative results as direct saliva. We also studied saliva added at 1:2 ratio in eNAT buffer, where a commercially available 2 ml eNAT media is added with 1 ml of saliva and showed better performance than nasal or oral swab. Thus, we do see the potential for a sample collection media which can keep the sample both stable and sterile for a period at different temperatures. 

Q:2. There is only a brief description of the Xpert Xpress SARS-CoV-2 test. Since this test was designed and validated for upper respiratory specimen such as nasopharyngeal swabs but is being applied to saliva specimen perhaps a more detailed explanation is appropriate. For instance, from what I could gather the volume of sample loaded into the Xpert Xpress cartridge is ~300uL. This is a significant deviation from standard RT-PCR protocols which use on the order of 20uL total for a single reaction.

Ans: Please see the above related answer to Q1. The application of Xpert Xpress SARS-CoV-2 test for different sample types is described in detail in another study published by some of the authors from this study, which is available as a pre-print in MedRXIV (Banada et al., https://doi.org/10.1101/2021.03.03.21251172)

Q: a. The authors only report results for the N2 target gene in the manuscript. This is certainly adequate to monitor for the presence of viral RNA, but is there a reason the results for the E target were omitted? Was there any interference from the eNAT on the amplification of the E target?

Ans: According to the FDA-EUA application of the Xpert Xpress SARS-CoV-2 test, the result of a confirmed positive test is made based on the N2 Ct values and not E alone. The test is confirmed SARS-CoV-2 positive when both N2 and E are positive or N2 positive and E negative. When E alone is positive, the test calls it presumptive positive for SARS-CoV-2. Thus, for the purpose of this study, only the N2, which is the signatory gene Ct was used. However, our analysis of E gene, showed both genes were detected in all positive samples, except for a single drop out in one of the replicates (1/4) on day 7 of 1:2 eNAT in saliva samples stored at 28°C.

1. Barra, G. B., T. H. Santa Rita, P. G. Mesquita, R. H. Jácomo and L. F. A. Nery (2021). "Overcoming Supply Shortage for SARS-CoV-2 Detection by RT-qPCR." Genes 12(1): 90.

2. Loeffelholz, M. J., D. Alland, S. M. Butler-Wu, U. Pandey, C. F. Perno, A. Nava, K. C. Carroll, H. Mostafa, E. Davies, A. McEwan, J. L. Rakeman, R. C. Fowler, J. M. Pawlotsky, S. Fourati, S. Banik, P. P. Banada, S. Swaminathan, S. Chakravorty, R. W. Kwiatkowski, V. C. Chu, J. Kop, R. Gaur, M. L. Y. Sin, D. Nguyen, S. Singh, N. Zhang and D. H. Persing (2020). "Multicenter Evaluation of the Cepheid Xpert Xpress SARS-CoV-2 Test." J Clin Microbiol 58(8).

---

## [Editor Report · Decision Letter 1]

20 May 2021

Inactivation of SARS-CoV-2 virus in saliva using a guanidium based transport medium suitable for RT-PCR diagnostic assays.

PONE-D-21-08255R1

Dear Dr. Banada,

We’re pleased to inform you that your manuscript has been judged scientifically suitable for publication and will be formally accepted for publication once it meets all outstanding technical requirements.

Kind regards,

Ruslan Kalendar, PhD

Academic Editor

PLOS ONE

---

## [Editor Report · Acceptance letter]

2 Jun 2021

PONE-D-21-08255R1 

Inactivation of SARS-CoV-2 virus in saliva using a guanidium based transport medium suitable for RT-PCR diagnostic assays. 

Dear Dr. Banada:

I'm pleased to inform you that your manuscript has been deemed suitable for publication in PLOS ONE. Congratulations! Your manuscript is now with our production department. 

Kind regards, 

on behalf of

Prof. Ruslan Kalendar 

Academic Editor

PLOS ONE